# Trait mediation explains decadal distributional shifts for a wide range of insect taxa

Yoann Bourhis [1] ✉, Alice E. Milne [1], Chris R. Shortall[1], Björn Beckman [2], Dan Blumgart[1], Rowan Edwards[3], Luke C. Evans[4,5], Chris W. Foster [4], Richard Fox [5], Marc S. Botham[6], Clare Rowland [7], Stuart Roberts[8], Martin C. D. Speight[9], Chris Hassall [10], William E. Kunin [10] & James R. Bell[11]

Shifts in insect distributions have been reported globally, largely attributed to climate and landscape changes. Communities are being reshaped, with species response traits mediating the effects of changing environments. Using a machine-learning approach we model 1252 insect occupancies across three decades in Great Britain. We combine independent models of nine insect groups (butterflies, moths, odonates, orthopterans, carabids, ladybirds, bees, wasps and hoverflies) to take a high-level view of the trends and key environmental drivers of insect occupancy, as well as to highlight the trait mediations underlying the resulting niches. Across this wide taxonomic range, we identify common trends in insect occupancies, showing no Great Britain-wide decline since 1990, but instead local declines and changes in community compositions. Known drivers of biodiversity loss appear to underlie those changes, notably urban sprawl and landscape simplification. Our approach also highlights the crucial roles of two response traits: habitat breadth, in mediating the effects of changing landscapes diversity and voltinism, in mediating the effects of increasing temperatures on insect life cycles.

Insect declines have been reported globally with varying magnitudes and consistencies across taxa. While declines in insect abundance have been widely observed[1–6], changes to occupancy[7–9] and more generally species richness[10,11] are increasingly reported. These changes have repercussions on other trophic levels, such as birds[12–14], and may undermine the sustainability of ecosystem services[15–17].

Threats to insects are complex and numerous[18], but there are two main categories of abiotic drivers behind patterns of biodiversity changes: landscape change and climate change. Landscape change includes general loss of natural areas aligned with encroaching urbanisation[19,20], as well as landscape simplification linked to agricultural management[21–24]. These constitute reductions in the quantity and quality of insect habitats, producing more homogeneous insect communities[25], with losses of specialists[26,27]. Unlike landscape change, climate change has less predictable impacts on insect distributions in the mid latitudes, despite the profound effect of temperature determining the rate of insect metabolism and development. Increasing temperatures benefit species at their poleward range margin, while

[1]Net Zero Resilience Farming, Rothamsted Research, West Common, Harpenden, UK. [2]British Trust for Ornithology, The Nunnery, Thetford, UK. [3]Hymettus Ltd, Midhurst, UK. [4]School of Biological Sciences, University of Reading, Whiteknights campus, Reading, UK. [5]Butterfly Conservation, East Lulworth, Wareham, UK. [6]UK Centre for Ecology and Hydrology, Crowmarsh Gifford, Wallingford, UK. [7]UK Centre for Ecology & Hydrology, Library Avenue, Bailrigg, Lancaster, UK. [8]School of Agriculture, Policy and Development, University of Reading, Reading, UK. [9]School of Natural Science, Trinity College, Dublin, Ireland. [10]School of Biology, University of Leeds, Leeds, UK. [11]School of Life Sciences, Keele University, Keele, Newcastle, UK. ✉e-mail: bourhis.yoann@gmail.com

negatively affecting others through disturbed seasonality and more frequent extreme weather events[28–30]. Depending on the spatial scale[31,32], habitat loss and climate change interact to affect insect distributional shifts over time, ultimately filtering communities into different compositions[8,33,34].

The overall reported declines hide large disparities among insects[35], with winners and losers possibly distributed along species trait gradients[36–38]. In such cases, it is reasonable to postulate that one or several species traits—e.g. body size, dispersal ability, or reproductive strategy—are modulating the effect of one or more environmental drivers[39]. Identifying these trait mediations and how they operate is therefore important in deciphering fundamental ecological processes that underly environmental filtering[40–44]. Conversely, environmental filtering may simply be species-specific, with marginal[45,46] or non-existant[47,48] trait mediation.

In this study, we build on the unparalleled insect occupancy data of the UK[49] to investigate the trends in a wide range of insect taxa (1252 species) across Great Britain over the period 1990–2020. There are a few modelling approaches capable of untangling trait-driver interactions[50,51]; here, we used an artificial neural networks (ANN) approach to explore trait-driver relationships in high dimensionality using an extensive trait database, as well as a large set of land cover and weather-derived drivers[52].

Here we train ANNs to predict occupancy of 1252 species spanning nine major insect groups (butterflies, moths, odonates, orthopterans, carabids, ladybirds, wasps, bees and hoverflies), building on a total 75 response traits and 30 environmental drivers. With this unique taxonomic breadth, we investigate the insect distributional shifts across Great Britain, and identify the leading environmental drivers behind those shifts, as well as the species response traits by which those drivers take effect. Finally, we explore the dominant local and regional drivers influencing insect distributional changes across Great Britain.

Our findings demonstrate that trait mediation plays a crucial role in providing a broad, taxonomically inclusive perspective on insect distributions. More specifically, we test the hypothesis that influential drivers are mediated in similar ways across groups such that insect-wide trait mediations can be identified, ultimately linking to fundamental physiological and ecological trait-mediated responses common to all insect groups.

## Results
### Changes in occupancy
Across the 1252 species, the occupancy records translate to a diversity of species-specific trends. However, the overall trends in the data (Fig. 1, blue curves) show some possible regional insect declines, notably in the north of Great Britain. Those records were used to train multi-species distribution models (see "Methods"), resulting in nine models (one per insect group) that produce species-specific prediction of probability of presence. The predictions at the locations and times of those observations result in very similar regional trends (Fig. 1,

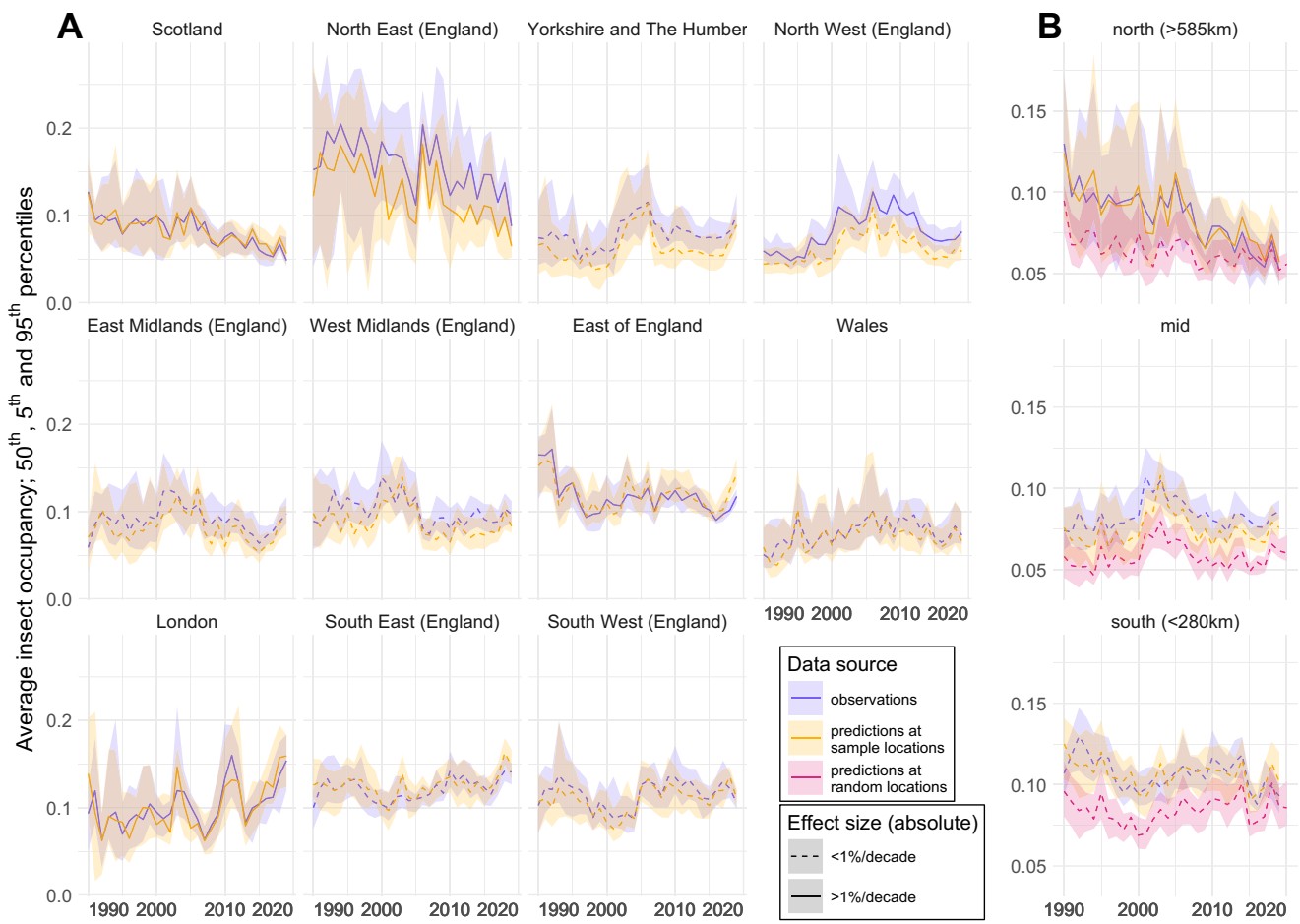

**Fig. 1 | Occupancy trends across 1990−2020. A** Observed (in blue) and predicted (at sample locations, in orange) average insect occupancy for the 11 ITL-1 regions of Great Britain. The uncertainty envelopes show the 90% prediction interval resulting from bootstrapped aggregation (*n* = 20 bags, see "Methods"). For both observations and predictions, only the test split of the dataset is shown. **B** Same but for 3 latitudinal slices of equal landmass along the length of Great Britain. Predictions at random locations are also displayed (in red), theoretically less affected by sampling biases. The solid lines mark the statistically significant (two-sided *t* test, *p* = 0.01, *n* = 600, adjusted for multiple comparisons testing with Bonferroni correction) linear trends with absolute effect size above 1% probability of presence per decade. Source data are provided as a Source Data file.

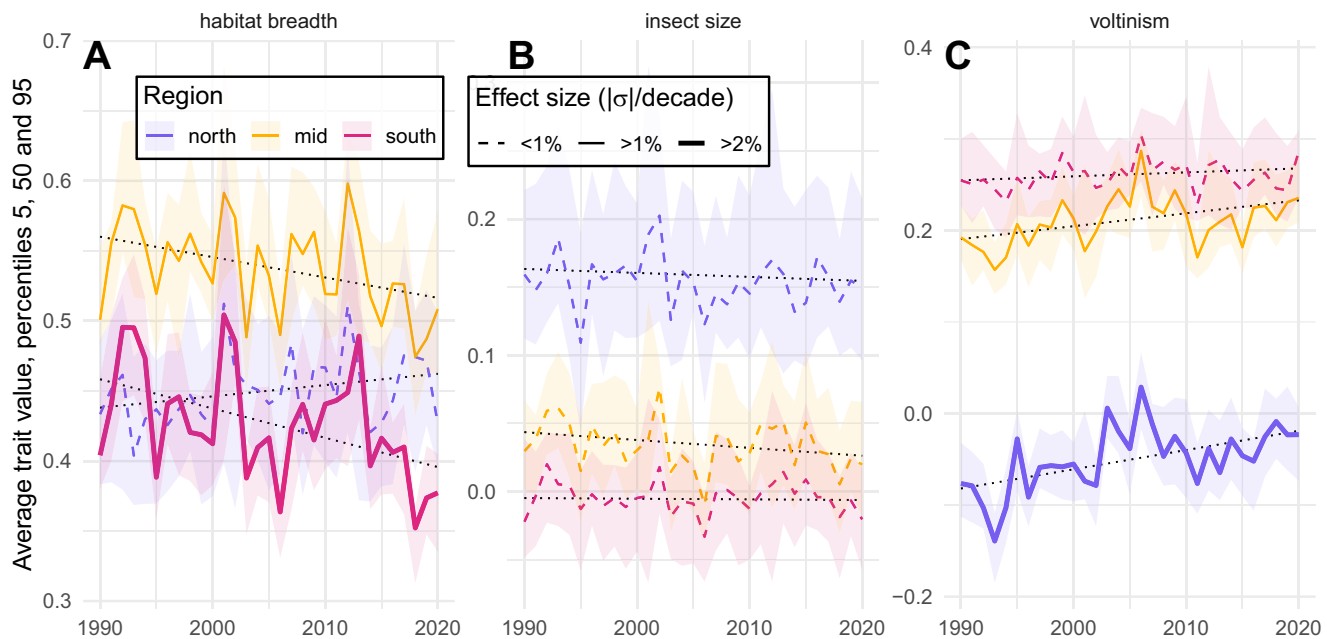

**Fig. 2 | Changes in trait assemblages.** Predicted average habitat breadth (**A**), insect size (**B**) and voltinism (**C**) across the 1990–2020. Predictions are produced at random locations across the three latitudinal slices of Great Britain (i.e. red curves in Fig. 1B). Uncertainty (90% prediction intervals) arises for the bagged models ($n = 20$). Ordinate units are standardised (centred reduced) trait values. The dotted black lines are included to show the direction of the linear trend. The solid lines mark the significant (two-sided $t$ test, $p = 0.01$, $n = 620$, adjusted for multiple comparisons testing with Bonferroni correction) linear trends with absolute effect size above 1% (or 2%) standard deviation ($\sigma$) of the trait value per decade. Source data are provided as a Source Data file.

orange curves). In both cases (blue and orange), these curves were derived from the samples reserved for the test datasets and not used for training (see Supplementary Fig. S2 and Supplementary Note 1 for a summary of model performances upon which our predictions are contingent). For ease of interpretation, we show observations and predictions split into three latitudinal slices of equal landmass (Fig. 1B), as we expected a strong latitudinal effect to appear. Figure 1B additionally includes predictions at locations that have been randomly selected (across a grid of Great Britain 1990–2020 with 1 km² · one year resolution) and are more representative of the regional changes (red curves). We note that the downward trends in the north (Scotland and Northeast of England) present significant effect size, despite the curves showing high uncertainties due to sample scarcity in Scotland, especially before 2007. However, predictions at random locations (red curve) are less strongly decreasing. We also note that, across Great Britain, the predictions at random locations present lower probability of presence than the predictions (and observations) at sampling locations. This could reflect a preferential sampling towards areas of preserved ecological interest by recorders.

### Shifting trait assemblages
Although trends in occupancy may appear limited to the north of Great Britain (Fig. 1B), this should not be interpreted as the absence of change since 1990 in the rest of the country. Most species have seen their occupancies retract, increase or shift. Even if no overall decline of insect occupancy is apparent, local communities are still likely to be modified. As a result, it is then possible that the local distribution of values for a given response trait is also shifted.

Most of the traits used in our models are taxa-specific, with very few common shared traits across insect groups. However, we selected three traits that have some equivalence and were documented for most taxa: voltinism, insect size and habitat breadth (i.e. *eurytopicness*). Voltinism (number of generations per year) and habitat breadth (number of different habitats the species can be found in) both have high equivalence across all taxa, while size covers different taxa-

specific metrics such as forewing length, intertegular distance or the whole insect length.

Building on those three cross-taxa traits, we translated our predictions of occupancy into predictions of average trait value in order to examine trends in trait assemblages (Fig. 2). While all three traits appear to follow strong latitudinal trends, temporal trends are less evident. We note nonetheless, an upward trend in voltinism that is especially marked as we go north (Fig. 2C), suggesting Great Britain has become relatively more conducive to species with multiple generations per year in comparison to low voltinism species. A decrease in habitat breadth in the south of Great Britain is also noted (Fig. 2A) as well as a lack of temporal trend in insect size (Fig. 2B).

### Drivers of changes
Using the Shapley Additive exPlanations (SHAP) method[53], the effect of every drivers on insect occupancy was estimated and summarised in Fig. 3. The drivers are detailed in Supplementary Table S1. The results suggest that the percentage of urban land cover has mostly a negative effect across taxa, but some groups (e.g. ladybirds and moths) respond more strongly and more coherently than others (e.g. carabids and bees). Largely positive effects of different metrics related to increasing temperatures (e.g. temperature diurnal range, maximum, or previous year average) are also observed. In Fig. 3, the drivers are highlighted depending on how consistently (see "Methods") they appear to have a mostly positive or negative effect on the focal insect group.

The role of trait-mediation in occupancy trends was also investigated. However, because the selected three common traits are expressed on taxa-specific scales, they are difficult to sensibly combine onto the same scale in order to derive a global assessment of trait mediation. To overcome this problem, we grouped the species into low, medium and high values of each trait at the insect group level (see Supplementary Fig. S4 for illustration), allowing us to make cross-taxa comparisons and hence explore insect-wide trait mediations.

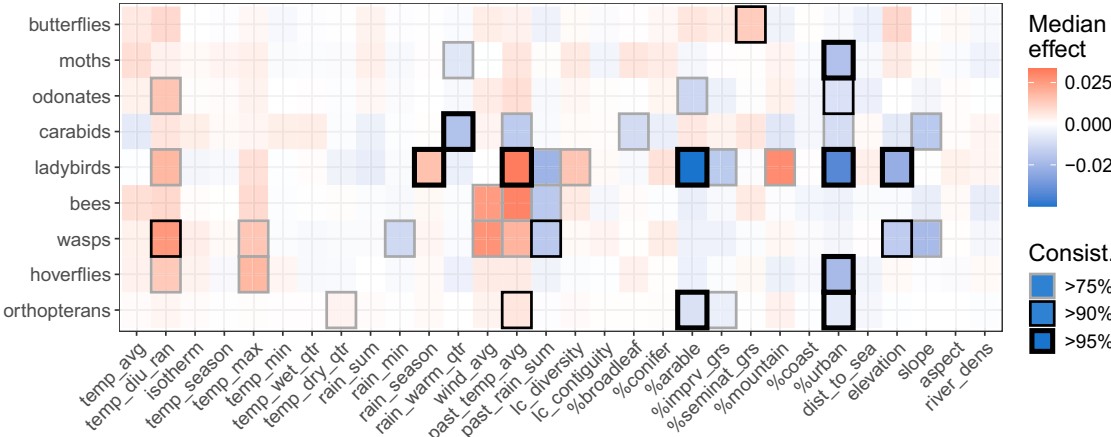

**Fig. 3 | Median effect of the different environmental drivers on the predicted probability of presence in the nine communities of interest.** Every pixel represents a distribution of effects, whose colour marks the median value (in red for positive effect and in blue for negative effect). The transparency and outline mark its consistency across bootstrapped replicates (i.e. bags), that is how consistently at least 75% of the distribution of effects falls above or below zero. Drivers are detailed in Supplementary Table S1. Source data are provided as a Source Data file.

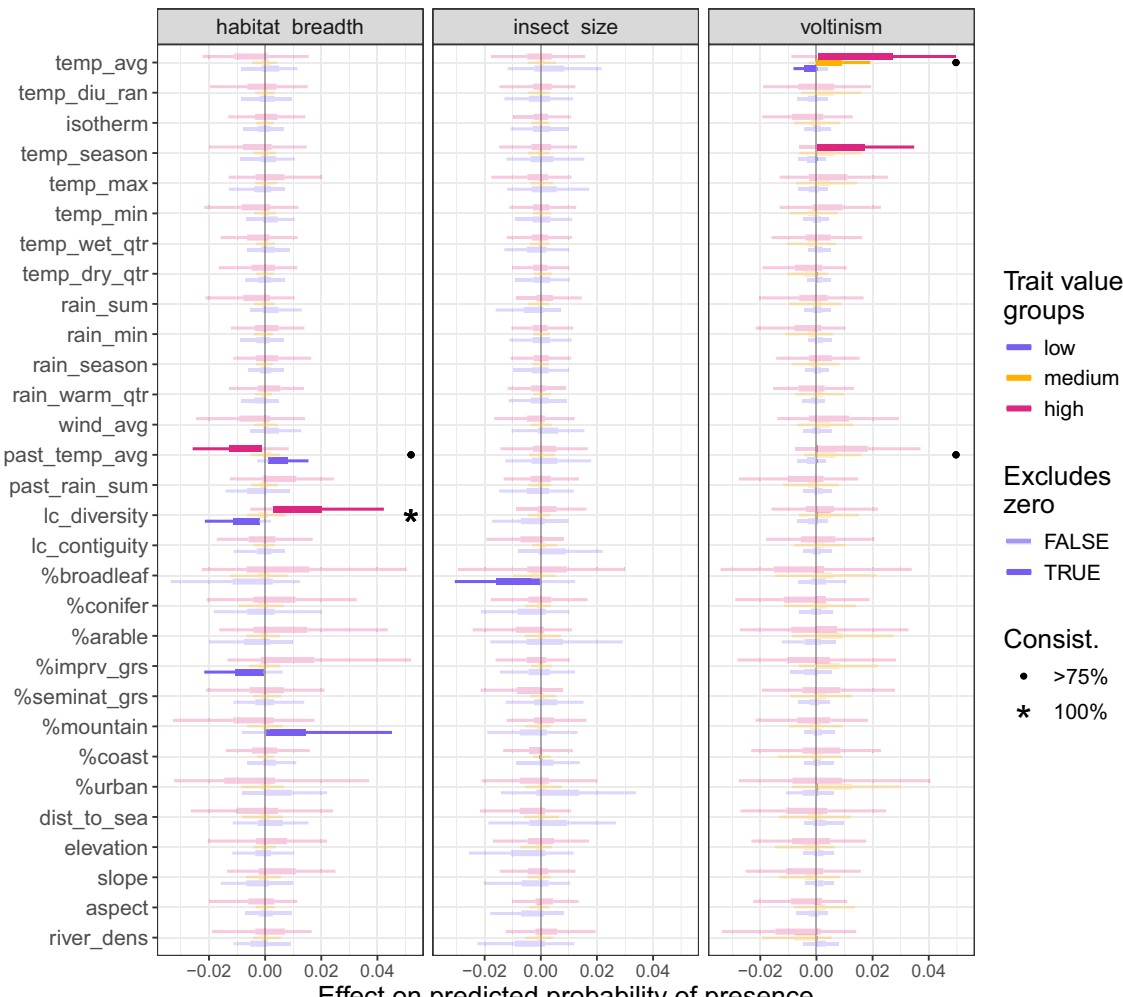

**Fig. 4 | Trait mediation of the environmental drivers.** For every driver, three boxes (made of the 5th, 25th, 50th, 75th, and 95th percentiles) show the distribution of effects of the focal driver for species grouped into low, mid and high value of the focal trait. Trait mediation appears as a differential driver effect along the trait value groups, with a strong trait mediation arbitrarily defined as an overlap of less than 25 percentiles between two of the groupings. The black dots mark such strong mediations that are consistently strong from bag to bag (i.e. in at least 15 of the 20 bootstrap replicates). Source data are provided as a Source Data file.

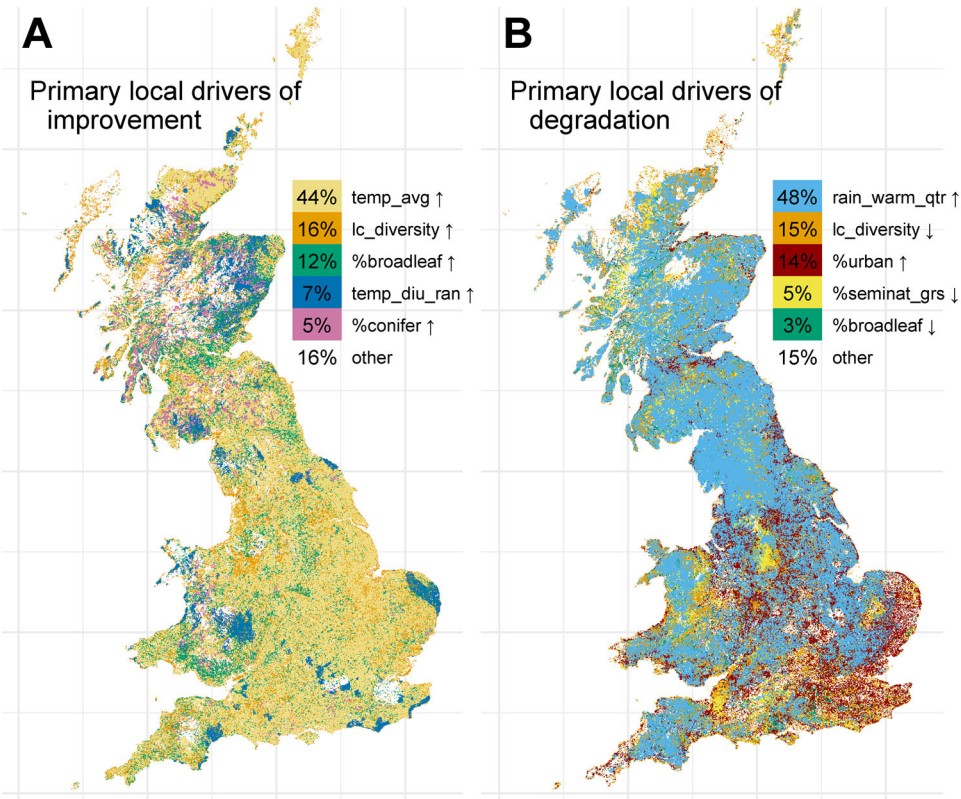

**Fig. 5 | Maps of the primary local drivers of insect occupancy changes.** The drivers are ordered by geographical coverage for which they are the key cause of increased predicted probability of presence (**A**) and decreased predicted probability of presence (**B**). The percentage values in the legend indicate the proportion of the map covered by each of the colours. For example, **B** shows that increases (↑) in the proportion of urban cover and decreases (↓) in the proportion of broadleaf forest are the first causes of degradation in 17% and 5% of the map respectively. See Supplementary Fig. S6 for the second and third leading causes of improvement and degradations. Source data are provided as a Source Data file.

In Fig. 4, trait mediation appears as a trend in the alignment of the three groupings (*low*, *medium* and *high*), where the percentile 25th–75th do not overlap between at least two of the trait groupings (i.e. the boxes in the boxplot). For example, in the panel *voltinism* we note one highlighted trait mediation: with *temperature average*. In this case, the high trait value group (in red) is associated with positive effects from increases in the *temperature average*, while the low trait value group (blue) is associated with mostly negative effects. A similar positive mediation is observed for the temperature average of the previous year (*past_temp_avg*). We also note consistent mediations by the *habitat breadth* trait of the land cover diversity (*lc_diversity*, positive), the proportion of improved grassland (*%imprv_grs*, positive) and the proportion of mountainous terrain (*%mountain*, negative). However, no sufficiently strong and consistent mediation by *insect size* is observed.

Having derived directional impacts of all assessed drivers on insect occupancy, we then mapped which drivers matter most locally. To this end, we derived a slope of change (across the period 1990–2020) for every driver at every 1 km × 1 km pixel of Great Britain using linear regression. Using our model explanations, environmental change was translated into differential impacts on predicted insect occupancy across Great Britain. From this approximation, the driver whose change mattered most locally was highlighted. Figure 5 shows the five most widespread drivers that are positively (Fig. 5A) and negatively (Fig. 5B) associated with insect occupancy. We observe for example that increasing average temperatures (temp_avg) drives increases in insect occupancy, particularly in eastern England. Whereas the increasing summer precipitation (rain_warm_qtr) is the most widespread first driver of decline, especially in the north.

## Discussion

Our model predictions show no overall insect occupancy decline in Great Britain since 1990. Yet, even if recent literature concurs with our results for some taxa (e.g. butterflies[33] and moths[54]), some taxonomic groups have been shown to retract (e.g. carabids, bees and hoverflies[7]). The apparent steady state of our overall predictions conceals changes in biodiversity. As they appear across taxa (see Supplementary Fig. S5 for group-level trends), these changes can also become visible through the prism of functional ecology (as shown by Neff et al. in Switzerland[38]), with occupancy trends that depend on species response traits. As hypothesised in the introduction, we were able to identify two key trait mediations that are consistent across the studied taxonomic groups. First, habitat breadth strongly mediates the effect of landscape diversity, and second, voltinism mediates the effect of temperatures.

According to our predictions, rising temperatures are the main cause of increasing average occupancy across the country (Fig. 5). This increase is mediated by insect voltinism (Fig. 4), resulting in shifts in community compositions towards a higher degree of multi-voltine species (Fig. 2). This result corroborates existing empirical findings that global warming expands the period during which multi-voltinism is possible[55], but raises concerns for uni-voltine species that may not adapt under a warming climate[56,57]. It also raises more theoretical concerns for the "winners" guild, with the additional generations allowed by a warmer climate possibly pushing populations into ecological traps (see the lost generation hypothesis[58]).

The observed decreasing occupancy in the north (Fig. 1) appears contrary to ecological expectations, as this is where the increasing temperatures should have the most positive impact due to possibly

unhindered range expansion from the south[56]. One possible explanation is that of a *shifting baseline* effect[59,60] wherein the species loss happened prior to the observation window in the south (i.e. before 1990), but within that window for the sampling sites in the north (i.e. since 1990). Note that when looking at our predictions at random locations in north in Fig. 1, the decrease is less strong, suggesting that the decrease chiefly concerns areas of preferential sampling. The north hosts most of the protected areas of the country, particularly extensive National Parks, and, as such, the unexpected decrease in insect occupancy echoes the strong biomass decrease that was recorded in protected areas of Germany over the same period[61]. This aligns with other evidences that poleward range expansion is a complex phenomenon, not only driven by the expansion of a species thermal range, but by a broader environmental filtering that also depends on land use[62] and species traits[56,63].

Another climate driven change is due to the amount of summer rainfall. Its increase over the period of interest appears to be the most widespread leading driver of decreasing average occupancy (Fig. 5). It is not connected to any of our tested traits (Fig. 4), possibly affecting insect species indiscriminately through more frequent extreme rainfall events[64]. However, little is known on the putative effect of rainfall on insect occupancy and its expected interactions with the behaviours of the recorders and insect activity.

As would be expected, urban cover shows a negative impact on insect occupancy (Fig. 3), especially in the south of Great Britain (Fig. 5), affecting strongly but without consistent trait mediation four of the groups (moths, ladybirds, hoverflies and orthopterans). We also note that, although the woodland habitat was not shown to strongly drive occupancy (Fig. 3), changes in woodland covers appear to dominate a significant portion of Great Britain (Fig. 5), possibly resulting in positive average effects across insects. As such, afforestation could appear as an effective tool to improve biodiversity with broad effect but, as warned by Bowler et al.[65], we suspect that a finer woodland classification and a wider set of traits would result in a more nuanced message. This is illustrated in a previous study where the positive effect from woodland on butterfly occupancy appears mediated by the hostplant type of the species[52], a trait not explored in our present broad trait analysis as comprehensive data is lacking for all but Lepidoptera.

Finally, the most clearly identified explanation is the trait mediation of landscape diversity by the habitat breadth (Fig. 4). This agrees with empirical findings that specialist species fare less well in highly diverse landscapes as, all other things being equal, such landscapes more likely offer them reduced habitat quantity and reduced connectivity[66–68]. In terms of trait assemblage, we can link reduction of the average habitat breadth in the south (Fig. 2) to the widespread loss of landscape diversity in this part of Great Britain (Fig. 5B). We note nonetheless that local landscape diversity has increased in many areas, and from these local trends an increased insect occupancy is expected (Fig. 5A). As such, we concur with the existing literature highlighting increasing landscape diversity as a relevant tool in preserving biodiversity[22,69,70]. Yet, to alleviate the possible negative effects of such increases on specialists, landscape management should aim to maintain a level of regional landscape heterogeneity by increasing landscape diversity in varied ways, while enhancing connectivity where possible[25].

Citizen science delivers robust, spatially-extensive data to assess range shifts, changes in habitat use and voltinism adaptations under a changing climate. Incorporating such decentralised observations along with data from expert-led monitoring schemes, our models support previous findings that climate change (particularly increasing temperature) and changing land cover (particularly urban sprawl) appear to drive the shifts we observe today in insect biodiversity at a broad scale. We were able to attribute heterogeneity within these overall trends in occupancy to a relatively small number of insect traits that are focused on developmental responses to temperature and breadth of habitat use. Our study, therefore, underlines the role local and national stakeholders can have in preserving biodiversity, notably by maintaining a regional heterogeneity of complex and diverse landscapes. However, considering the greater sensitivity of uni-voltine species to increasing temperatures, offsetting climate change through landscape management only will invariably result in changed insect communities.

## Methods
### Data
Three types of data were used for our modelling approach. These are occupancy data (our dependent variable), as well as environmental drivers and species response traits (our independent variables).

The occupancy data are species observations from across Great Britain made during the period 1990–2019. The data for the butterfly and the moth groups are sourced from structured recording schemes (from the UK Butterfly Monitoring Scheme and the Rothamsted Insect Survey respectively). Those are *full list* records in which all the species of the group are looked for during extensive and standardised recording events. They can effectively be regarded as presence-absence data sets, even though their absences inherently come with varying levels of uncertainty. The remaining datasets result mostly from single species observations (from the NBN atlas) from which were derived presence-absence contingency tables[71] at a $1 km^2$–one year resolution. This caused an artificial zero-inflation that was compensated for with species-specific class weights[52,72,73] that lower the importance of those artificial absences (zeros) during learning, hence diminishing their negative effects while enabling their processing through our ANN. See the sample distribution in Supplementary Fig. S1.

The environmental drivers comprise weather and land cover metrics which are detailed in Supplementary Table S1. They originate from two main sources: the Had-UK grid[74] from which was derived the BIOCLIM19 set of metrics, and UKCEH Land cover data[75–80] from which were derived compositional metrics, as well as the Shannon diversity[81] and spatial contiguity[82] indices of the land cover types. All metrics are informed yearly at the $1 km^2$ resolution. The weather metrics are informed yearly, while the landscape metrics are linearly interpolated between the years in which they are informed (i.e. 1990, 1994, 1998, 2002, 2006, 2010, 2015, 2017, 2018, 2019 and 2020). Additionally, we included five static topological drivers: elevation, slope, aspect (all three from AWS via elevatr), distance to sea, and density of the river network[83]. The inception of the high quality, Great Britain-wide land-cover data in 1990 is the key determinant for the starting year of our period of interest.

Biological response trait data were sourced from both published[84–86] and unpublished sources. For groups where traits were available, we only retained the species for which the full set of selected traits was informed, and for which there is a minimum of 30 observations across the period of interest. This amounts to 1252 species (58 butterflies, 435 moths, 41 odonates, 23 orthopterans, 115 carabids, 45 ladybirds, 190 bees, 151 wasps and 194 hoverflies). Note that wasps were addressed without trait data, as none were found. In addition to response traits, all communities were given a *phylogeny* trait in the form of a vectorised (i.e. flattened) taxonomic tree. This additional *trait* can help model performance by carrying hypothetical mediations by missing traits that align strongly with taxonomy, as is common in other multi-species distribution models[50]. Note that because we lacked trait measurements of the same nature across insect groups, especially for insect sizes, we did not investigate their absolute insect-wide effects. Instead, those measurements were scaled within each group (see Supplementary Fig. S4) so that we could highlight within-group effects that were consistent at the wider insect level.

## Modelling

Uncovering trait mediations is challenging with existing methods, as testing for trait-driver interactions often relies on hierarchical Bayesian models[50,87,88] that do not scale well to high dimensionality. Here, we used a recently described method[52] that builds on artificial neural networks (ANN) to remain tractable while exploring high numbers of samples, traits and drivers. See Supplementary Fig. S2 and Supplementary Note 1 for details on model selection and performances.

The approach uses ensemble of two ANNs whose predictions are averaged (i.e. the *ensemble* architecture described in Bourhis et al. 2023[52]). The first ANN is trained to learn a shared response, which is a response to the environmental drivers that applies to the whole community as a function of the individual trait values of the species – this is where trait-mediation occurs. The second ANN is trained to learn the species-specific direct response to the drivers. While this second ANN is very flexible, the former is quite rigid as it is trying to learn parameters that apply to the entire community. However, as this rigidity results in learning trait mediation, this drastically improves the performance for poorly informed species with few records, as for such species, it assumes stereotypical responses dictated by species' traits.

Our models were *bagged* (bootstrap aggregation[89]) in order to quantify epistemic (data) uncertainty. Essentially, instead of one community model, an ensemble of models is produced in which constituent submodels are trained with varying (bootstrapped) input data. From one bag to the next, only half the data is in common. This approach results in varying predictions from which dispersions and central tendencies can be derived (here the 5th, 50th, and 95th percentiles for our occupancy predictions, as well as the 25th and 75th additionally for our model explanations). Bagging reduces the chances that our predictions, and their explanations, become unduly influenced by fluctuations in situations where information is scarce, making our conclusions more conservative. By making poorly sampled regions, periods or taxa more uncertain, bagging also attenuates the effects of sampling biases which are always a concern with species records, even with structured recording schemes[90–92]. See Supplementary Fig. S3 for a schematic view of the ensembling process.

Presence-absence data is naturally unbalanced and this needs to be taken into account for the learning phase not to result in a trivial model (e.g. a model predicting only absences for a rare species, disregarding the predictors). We applied species-specific class weights that enforce the same importance to the presences and the absences of a species. Put simply, if there are 10 times more absences than presences, each of them matters 10 times less than each presence during learning. For two classes like here (presence and absence, noted 1 and 0), the weight of a sample of each class ($w_0$ and $w_1$) depends on the number of samples in each class ($n_0$ and $n_1$) and is given by:

$$w_0 = \frac{n_0 + n_1}{2n_0}; w_1 = \frac{n_0 + n_1}{2n_1} \tag{1}$$

This defines linear class weights, and applying a square root to those defines square root class weights. We have found that the former work best for presence-only data while the later are best for presence-absence data (see Supplementary Fig. S2 and Supplementary Note 1 for further details).

Our predictions were produced across the whole of Great Britain and from 1990 to 2020. From those, we derived yearly average occupancy for all species considered. First, we compare the occupancy trends in the observation data with our predictions at the same sampling locations (both from the test data set, i.e. excluding training data), to check trend agreement between observations and predictions (Fig. 1). Then, to circumvent the sampling bias from the data, we investigate the predicted trends at random locations across Great Britain. To produce comparable uncertainties for the observed and predicted trends, the number of random locations was set per insect group and year to match the number of samples in the (test) data set for that year and insect group. Because we expected strong latitudinal structure in our results, we divided Great Britain into three slices of equal land area, resulting from latitudinal cuts at northing 280 km and 585 km of the British National Grid.

Finally, we used our species-specific predictions to investigate possible trends in the trait assemblages that they produce when combined. The change in trait assemblage resulted from the weighted averages of the trait's values, following Eq. (2)

$$\bar{t}_y^r = \sum_x^X \frac{\sum_s^S t_s . p_{s,x} . \max\{MCC_s, 0\}}{\sum_s^S p_{s,x} . \max\{MCC_s, 0\}} / |X| \tag{2}$$

with $\bar{t}$ being the trait average for year $y$ and region $r$, $X$ being the set of samples belonging to year $y$ and region $r$; $t_s$ is the trait value of species $s$, while $p_{s,x}$ is its predicted probability of presence at sample $x$. Finally, $MCC_s$ is the Matthew's Correlation Coefficient for species $s$, a measure of the model performance at predicting this species accurately[93]. This metric is particularly adapted to training SDMs as it is robust to the severe class imbalances inherently present in occupancy data sets. Supplementary Fig. S5 shows how the changes in trait assemblage result from the distribution of trends in average occupancy.

## Explanations

We used the Shapley Additive exPlanations (SHAP) method[53] to identify the driving factors behind our predictions, which is key to deriving understanding from any form of SDM. SHAP is a variable importance method that interprets machine learning models from local (i.e. sample-level) explanations. For a number of random samples (here, n = 50), a SHAP value is derived for every driver and every species. This SHAP value represents the effect of the driver on the response variable, which here is the probability of presence of every individual species. It is derived by subtracting the local predicted response by the ones derived with the focal driver being changed to several randomly picked background values (here, n = 50). Then for every species and every driver, we can linearly regress those SHAP values to the driver values, resulting in a linear slope coefficient that gives a linear approximation of the effect of the driver. With this method we derived the direct effect of every driver on every species, as well as the trait-mediated effect of every driver as a function of the species trait values. Like our predictions, these model explanations resulted from the bagged models and are therefore provided with uncertainty estimates.

Here we arbitrarily set that a direct driver effect whose distribution excluded zero by at least three quartiles was strong enough for scrutiny, and was marked as consistently strong if that exclusion was found in a sufficient number of bags. We set these thresholds of consistency to 15, 18, 19 and 20 bags (out of 20), marked in Figs. 4 and 5 as 75%, 90%, 95% and 100% consistency. A trait mediation effect was highlighted under the same conditions, but rather than excluding zero by three quartiles, two of the trait value groups (low, medium or high) needed to exclude each other by three quartiles.

## Local causes of shift

Over the period 1990–2020, the environmental drivers exhibited local changes of different magnitudes. We quantified this by identifying a linear slope coefficient for every driver at every 1 km$^2$ pixel of Great Britain. Then, multiplying this rate of change with the associated driver effect on insect occupancy (i.e. SHAP values), we ranked the local causes of putative shift in insect distribution. We then produced maps of Great Britain showing the most important local causes of increased and decreased occupancy. Additionally, Supplementary Fig. S6 presents for the second and third leading causes of improvement and degradations.

## Addressing sampling biases

Working across several taxa requires using occupancy data that involves varying recording efforts, taxonomic breadths and spatiotemporal coverages. In our case, it also involved using presence-absence and presence-only data jointly. This raises three types of concerns about sampling biases pervading our predictions and rendering them unfounded.

The first concern regards the use of presence-only data as presence-absence when building the contingency tables that the ANNs trains on. Doing so results in a great quantity of artificial zeros (i.e. pseudo-absences) that, for the relevant insect groups (here all but moths and butterflies), blurs the meaning of absence. The effect is a predicted probability of presence with a lower bound that holds a slightly different meaning than the one resulting from presence-absence data. While this allows us to compare and merge spatio-temporal trends across many insect groups, comparing absolute values between those groups remains unwarranted (and is not done here). An alternative approach would be the composition of a background data set with pseudo-absences generated at random locations[94]. However, there is support for instead simply using the presences of other species within the group as pseudo-absences for the focal species (what is done here), as doing so provides background samples with similar bias as the presence-only records[95,96].

The second concern regards spatiotemporal sampling biases (for which assessment tools exist[97]). For example, all the insect groups modelled here have the bulk of their observations made in southern Great Britain, with observations becoming rarer towards the north. It also appears that there are more observations reported in the last decade than in the previous two, as reporting has been made simpler through tools like *iRecord* and *iNaturalist*. The key concern is that trends in the quantity of reported information might weaken, amplify or even invert true trends in occupancy. It is therefore critical that predictions are made with proper uncertainty estimation, able to reflect the local scarcity of information (local in the whole input space, not only geographical space). This is here done with "bagging" ("bootstrap aggregating"), which is commonly used with machine learning approaches such as ours. Yet addressing biases properly remains an open question[94,98–102] and a noteworthy practice consists in sub-sampling (aka filtering or thinning) the data set towards more spatiotemporal balance[103]. We concur that without uncertainty estimation, such is best practice, but argue that it is here rendered redundant through "bagging".

The third concern is taxonomic bias, where not all species are given equal sampling efforts for a variety of reasons. This amplifies an already huge natural class imbalance, where e.g. observations of rare species become drowned in a great quantity of (pseudo-)absences. In such conditions, the use of species-specific class weights is needed for model training[52,73]. The cost of doing so is a drastic reduction of the effective sample size, that results in increased uncertainty. This is another reason why our predictions, and subsequent highlights, are only valid when provided with uncertainty estimates. Supplementary Note 2 provides an illustration (Supplementary Fig. S7) of how bagging and species-specific class weights work in synergy to address those three concerns in a simulated data experiment.

## Reporting summary

Further information on research design is available in the Nature Portfolio Reporting Summary linked to this article.

## Data availability

Occupancy data can be queried by contacting the respective monitoring schemes and societies for curated data sets: UKBMS for butterflies, RIS (https://www.rothamsted.ac.uk/national-capability/the-insect-survey) for moths, BDS (https://british-dragonflies.org.uk/recording/monitoring/) for odonates, BWARS for bees and wasps, or downloaded from the NBN atlas (https://nbnatlas.org/) for more diverse data collections. For drivers, land-cover data can be queried from UKCEH while weather data (Had-UK grid) can be queried from the CEDA archive (https://catalogue.ceda.ac.uk/uuid/4dc8450d889a491ebb20e724debe2dfb/). Unpublished trait data can be requested from Dan Blumgart (dblumgart@outlook.com) for moths, Björn Beckman (bjornbeckmann@mailbox.org) for orthoperans, James R. Bell (j.r.bell@keele.ac.uk) for carabids, Helen E. Roy (hele@ceh.ac.uk) for ladybirds, Stuart Roberts (spmr@msn.com) for bees and Martin C. D. Speight (speightm@gmail.com) for hoverflies. Source data are provided with this paper.

## Code availability

The python and R code used to produce our results is available[104] at https://zenodo.org/records/15572921. However, as most occupancy, driver and trait data are not free to share, we can only attach said data for the moth group as a minimal working example.

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

## Acknowledgements

This study is part of NE/V00686X/1 Drivers and Repercussions of UK Insect Declines (DRUID). We thank all the volunteers who helped in building up the huge amount of data used here. The Rothamsted Insect Survey, a National Bioscience Research Infrastructure, is funded by the Biotechnology and Biological Sciences Research Council under the award BBS/E/RH/23NB0006. The authors are grateful for the use of Open Data from the British Dragonfly Society Recording Scheme, available from NBN Atlas as data resource dr731. Thanks to the Bees Wasps and Ants Recording Society for species observation data. Thanks to C. Carvell and G. Powney for constructive criticism of the early manuscript. The UK Butterfly Monitoring Scheme is organised and funded by Butterfly Conservation, UK Centre for Ecology & Hydrology, British Trust for Ornithology and Joint Nature Conservation Committee. We thank Helen Roy from the Ladybird Survey for the occurrence and traits data.

## Author contributions

Y.B., A.E.M., and J.R.B. conceived the study. C.R.S., B.B., D.B., R.E., L.C.E., C.W.F., R.F., M.S.B., C.R., S.R., M.C.D.S. collected, extracted, and curated the data. A.E.M., J.R.B., W.E.K. and C.H. acquired the funding. Y.B. analysed the data (under the supervision of A.E.M. and J.R.B.). Y.B., A.E.M., J.R.B., and W.E.K. interpreted the results and wrote the first draft. All authors commented on the manuscript and approved the final version before submission.

## Competing interests

The authors declare no competing interests.
