## [Transparent Peer Review file · Nature Communications]

Trait mediation explains decadal distributional shifts for a wide range of insect taxa

Corresponding Author: Dr Yoann Bourhis

Version 0:

Reviewer comments:

Reviewer #1

(Remarks to the Author)

• What are the noteworthy results?

The authors use a unique 30-years data set on habitat occupancy of nine insect taxa across the UK to reveal temporal occupancy trends. They further compiled trait data for all (except one) insect taxa and link spatiotemporal trends to a range of environmental parameters related to land use and climate. The results indicate that there is no general decline in occupancy across the UK, yet for northern England and Scotland occupancy declined. Very interesting are the findings that average habitat breadth declined in the south of UK whereas voltinism, the number of generations per year increased in the north. Patterns for drivers of change are mixed but indicate increased probability of presence with warmer temperatures and decreased probabilities with higher % urban area.

• Will the work be of significance to the field and related fields? How does it compare to the established literature? If the work is not original, please provide relevant references.

The study is an interesting and relevant contribution to an increasing body of publications on long-term trends of insect declines, possible drivers, and interactions among climate and land use. In addition to references cited in the introduction, recent studies demonstrate temporal declines in insect biomass and abundance (Hallmann et al. 2017, Seibold et al. 2019), land use and extreme weather events as potential drivers (Seibold et al. 2019, Müller et al. 2023), and climate-land use interactions (Uhler et al. 2021, Ganuza et al. 2022). These studies might help the authors to provide a more differentiated picture of possible drivers and interactions in the introduction and discussion of own findings.

• Does the work support the conclusions and claims, or is additional evidence needed?

In principle, yes. However, the discussion is very short and general and lacks a more causal interpretation of findings and consideration of partly conflicting other studies. I also find the introduction too short and superficial and miss the development of testable hypotheses.

• Are there any flaws in the data analysis, interpretation and conclusions? Do these prohibit publication or require revision? Data analysis and interpretation mainly look sounds (but see my comment below). The decline in probability of occupancy in northern England and Scotland is actually contrary to expectations, given the positive impact of temperature and the decadal increase in temperature, but this is not discussed.

• Is the methodology sound? Does the work meet the expected standards in your field?

The authors use an artificial neuronal networks approach to explore the high number of samples, traits and potential drivers. Without own experience with this method I can't judge how valid the analyses are, and it is difficult to reconstruct which and how analytical steps have been done by the AI. Common shortcomings of such data sets such as spatial and temporal bias or gaps in sampling data and collinearity of environmental factors have been addressed in the data analyses but it remains unclear how critical these aspects are and to which extent they might question the quality of presented findings.

• Is there enough detail provided in the methods for the work to be reproduced?

No, I also missed basic information on the total number of occupancy data for each of the nine taxa and how these data are distributed in time and space. Most occurrence, trait and driver data are under embargo according to the authors and will be not published with this manuscript which is unfortunate

References

Ganuza, C., Redlich, S., Uhler, J., Tobisch, C., Rojas-Botero, S., Peters, M.K., Zhang, J., Benjamin, C.S., Englmeier, J., Ewald, J., Fricke, U., Haensel, M., Kollmann, J., et al. (2022). Interactive effects of climate and land use on pollinator diversity differ among taxa and scales. *Science Advances*. 8, eabm9359. <https://doi.org/10.1126/sciadv.abm9359>
Hallmann, C.A., Sorg, M., Jongejans, E. Et al. (2017). More than 75 percent decline over 27 years in total flying insect

biomass in protected areas. PlosOne, <https://doi.org/10.1371/journal.pone.0185809>

Müller, J., Hothorn, T., Yuan, Y., Seibold, S., Mitesser, O., Rothacher, J., Freund, J., Wild, C., Wolz, M., & Menzel, A. (2023). Weather explains the decline and rise of insect biomass over 34 years. *Nature*. <https://doi.org/10.1038/s41586-023-06402-z>

Seibold, S., Gossner, M. M., Simons, N. K., Bluethgen, N., Müller, J., Ambarli, D., Ammer, C., Bauhus, J., Fischer, M., Habel, J. C., Linsenmair, K. E., Nauss, T., Penone, C., Prati, D., Schall, P., Schulze, E.-D., Vogt, J. et al. (2019). Arthropod decline in grasslands and forests is associated with landscape-level drivers. *Nature*, 574. <https://doi.org/10.1038/s41586-019-1684-3>

Uhler, J., Redlich, S., Zhang, J., Hothorn, T., Tobisch, C., Ewald, J., Thorn, S., Seibold, S., Mitesser, O., Morini, J., & al. (2021). Relationship of insect biomass and richness with land use along a climate gradient. *Nature Communications*, 12. <https://www.nature.com/articles/s41467-021-26181->

(Remarks on code availability)

Reviewer #2

(Remarks to the Author)

This is my first time reviewing "Trait mediation explains decadal distributional shifts for a wide range of insect taxa" submitted by Bourhis et al. to *Nature Communications*. The text regards distributional shifts and the decline of insect species in Great Britain. Specifically, the authors considered 1252 insect occupancies in Great Britain in three decades from nine insect groups. The authors indicate known drivers of biodiversity loss to insect species based on trait mediation. The authors also considered species traits to determine the potential change of insect data across decades. The authors analyzed general variables affecting the amount of data for each group of species. For instance, the authors found that urbanization variables negatively affected the insect data while temperature verage positively fected the species. In general, the text is well written and below I provide some suggestions of improvement that need to be considered before the text is accepted for publication.

L60-63: Please note that "wider" and "higher" require a complete comparison. To what are you comparing "wider" and "higher" here? (e.g., "wider than XXX" and "higher than YYY").

L84: Please avoid using possessives in academic texts. It is too colloquial.

L91: "nine" - Please note that numerals lower than 10 need to be written in full lenght. On the contrary, numerals higher than 10 need to be written as numerals. Still, please avoid starting a sentence with a raw numeral.

L112: Please use en dashes in numerical ranges and "one year"

L118: Please explain the acronym "GB" here. Does it stand for "Great Brittain"? THis is the first time it appears without any other explanation.

L127: Please avoid using possessives in academic texts. It is too colloquial.

L190: Please use en dashes in numerical ranges.

L201: Please avoid using possessives in academic texts. It is too colloquial.

L209-210: "1 km x 1 km"

L261: Please avoid using possessives in academic texts. It is too colloquial.

L299: Please use en dashes in numerical ranges.

L316: Please note that if you are using the American English variant, you need to use a serial comma before "2020". Of you are using the British English variant, then you do not need it. Still, there are other instances where you used serial commas, for instance, L324.

L335: Please use commas after "here"

L341: Please avoid using possessives in academic texts. It is too colloquial.

L348-349: Please invert this sentence.

L353: American version (serial comma required) or British English (no serial comma required)?

L385: "We used the Shapley Additive exPlanations (SHAP) method"

L389 and 394: "n = 50"

L396: Please avoid using possessives in academic texts. It is too colloquial.

L405: American version (serial comma required) or British English (no serial comma required)?

L411: Please use en dashes in numerical ranges.

(Remarks on code availability)

Reviewer #3

(Remarks to the Author)

In this manuscript, the authors devised a deep neural network model for species distribution modelling (SDM) and predicted distributions of British insects by training it with large scale occurrence records and trait databases. The neural network model had a specific architecture to incorporate trait mediations (trait-environment interaction) terms on top of the basic prediction model with environmental variables. The authors tested the effects of trait mediation by using the model and a variable importance measure developed on the SHAP value, a popular metric for machine learning model explanation.

The most methodological parts of this manuscript were presented in Bourhis et al. 2023 (published in *Method in Ecology and Evolution*). This study is an application of the method to broader datasets. Having read through the 2023 paper, I find most concerns and questions I had during the review have been already addressed. The models and training procedures

are carefully designed and tested, and I do not find serious flaws in the methods such as data leakage. General concerns associated with SDM studies like class imbalance and biases of presence-only data were also properly addressed in this manuscript.

Therefore, the main results, like local decline of insects and the significant interactions between voltinism and temperature, are convincing. I feel that this study is a good application of the explainable AI, which can not only predict the species distributions but also detect important drivers of the decline.

Only one methodological aspect I think to be added is the test of importance measures. If I understand it correctly, the SHAP-based importance measure used in this study is a novel method proposed in the 2023 paper, and it has been tested only with the butterfly and moth data sets. The same testing procedure like removing unimportant features and checking performance reduction may strengthen the method's applicability.

Apart from the methodology, one problem of this manuscript is it lacks detailed descriptions of the methods used. The current method section is not sufficient for reproducing the study. Probably, the method section requires more specific details of the models, training procedure and whether there are differences between this study and the 2023 paper. For example, Bourhis et al. 2023 proposed and tested multiple architectures and training procedures. Which ones were used in this study?

In addition to more specific descriptions of models, schematic figures like the ones in the 2023 paper may help readers to capture the overview of the model.

Minor points:

Line 100: This study's results strongly depend on the performance of the model's prediction. Please briefly report the model's performance metrics, maybe MCC?, early in the result section.

Line 199: The effects of insect body size may act on the absolute values and may be erased when the relative size class is used to aggregate the effects on multiple taxa.

Line 366: How many locations is "a set number of random locations"?

Line 419 onward: I am generally convinced by the arguments here and Appendix D, but this section lacks some contexts. Can you briefly describe the common approaches used to correct the biases in the SDM applications, and how the current approach compares to them?

Line 451: What type of weighting is used in this study? Since the weighting strategy is critical to the model's good performance, more specific information should be provided. Can you write it down as an equation?

Gillespie et al. 2024 proposed a loss function which could control the similar biases by introducing a different weighting scheme. There may be some commonality between two approaches. (<https://www.pnas.org/doi/10.1073/pnas.2318296121>)

Line 721 (Appendix D): Is this simulation based on previous studies? If so, please cite them.

(Remarks on code availability)

The repository contains a brief instruction and all codes required for reproducing this study, including models, training procedures and visualization. You can reproduce the workflow with a decent computer and a typical environment for running deep learning models. I could run the code on my workstation but could not on my laptop because of high resource requirement.

Version 1:

Reviewer comments:

Reviewer #1

(Remarks to the Author)

I am reviewing this manuscript the second time. The revised version has taken into account my previous comments. The introduction, results and discussion are now clearer and more differentiated. I can accept that a more nuanced discussion of findings is not feasible in this format and see the value of such long-term and multi-taxa synthesis work.

The additional information on methods and modelling in the supplementary material is very helpful.

(Remarks on code availability)

Reviewer #2

(Remarks to the Author)

This is my second time reviewing "Trait mediation explains decadal distributional shifts for a wide range of insect taxa"

submitted by Bourhis et al. to Nature Communications. The first draft was already pretty decent and I see no major problems with this new improved version. After Reading it again, I raised some very minor issues shown below.

L34: "(e.g. butterflies)"...

L43: "landscape diveristy and voltinism" (no semi-colon).

Please check whether the jornal requires the use of keywords at the end of your abstract (or not).

General input to your intro: Considering the suggestion made by Reviewer #1 regarding the inclusion of some key references (e.g. Hallman et al. 2017), I believe you need to include other key references in your main text (intro and discussion). They are directly related to the topics presented in your study.

Dirzo, R., Young, H. S., Galetti, M., Ceballos, G., Isaac, N. J. B., & Collen, B. (2014). Defaunation in the Anthropocene. *Science*, 345, 401–406.

<https://www.science.org/doi/10.1126/science.adp4671>

Regarding landscape change, this new study is a must-be-cited one: <https://www.nature.com/articles/s41586-025-08688-7>

Regarding climate change, the study by Pecl et al. 2017 needs to be cited as well.

Pecl, G. T., Araújo, M. B., Bell, J. D., Blanchard, J., Bonebrake, T. C., Chen, I.-C., Clark, T. D., Colwell, R. K., Danielsen, F., Evengård, B., Falconi, L., Ferrier, S., Frusher, S., Garcia, R. A., Griffis, R. B., Hobday, A. J., Janion-Scheepers, C., Jarzyna, M. A., Jennings, S., ... Williams, S. E. (2017). Biodiversity redistribution under climate change: Impacts on ecosystems and human well-being. *Science*, 355, eaai9214. <https://doi.org/10.1126/science.aai9214>

L97: "groups (e.g. butterflies,)"

L102: "mapped"

L110: please remove this line.

L114: "translated"

L117: "were used to train"

L120: "resulted" (please use the simple past tense throughout the entire Results section, as this tense is often indicated to be used in this kind of section).

L184, L185, and L186: "(e.g. ladybugs" and "(e.g. carabids and bees)" and "e.g. temperature", respectively (since you appear to be using the British English variant in your text).

Figure #5: Could you consider a different color palette here? Blue and green are pretty close and difficult to see. I think there are R packages capable of allowing you to color your figures in more easy-to-see colors. Please check this (at least for Fig5A).

L245: "showed" (please also use the simple past tense in all verbs related to the results you are presenting in the Discussion section).

L323: PLease avoid possessives in academic texts. It is too colloquial.

L340-341: Considering that there are vast available biological information for these groups in online datasets like GBIF and other museums and considering the baseline shift hypotheses you proposed as one of the explanations of your results in the discussion, please explain to your readers why you did not use GBIF data with dates prior to 1990 here.

For the discussion, since you used iRecord and iNaturalist data, I think it would be interesting to call your readers' attention regarding the importance of citizen science to help develop analyses like ours.

Please use the simple past tense in your M&M section as well, when you explain the methods you used to reach determined results.

L369: "species (e.g. 58 butterflies,)"...

L408: Please, but please avoid using the "to get" verb and all of its variants in academic texts. It is too colloquial!

(Remarks on code availability)

Reviewer #3

(Remarks to the Author)

This manuscript is a revised version of the manuscript I reviewed. It reports insect distribution shifts and their major drivers predicted by a neural network model trained with long-term occurrence datasets.

Authors thoroughly responded the points raised by the reviewers. Additional texts and figures are informative, especially, Fig S1 and S2 provide details of model evaluation and model overviews. Also, the difference between this work and Bourhis et al. 2023 is clearer in the current version. I think now the manuscript and codes provide sufficient details for ensuring evaluation and reproduction of this research. As I already mentioned in the initial review, the manuscript's results are convincing and now they are better discussed.

Overall, I think the manuscript is now in very good quality and ready for publication.

There is only one minor point to be added:

Line 524: how about adding non-abbreviated form ("bootstrap aggregating") after "bagging"?

(Remarks on code availability)

It is the same repository as the previous review, but with a smaller dataset for better accessibility. All codes for analyses and visualisation are included.

REVIEWER COMMENTS

Reviewer #1 (Remarks to the Author):

- What are the noteworthy results?

The authors use a unique 30-years data set on habitat occupancy of nine insect taxa across the UK to reveal temporal occupancy trends. They further compiled trait data for all (except one) insect taxa and link spatiotemporal trends to a range of environmental parameters related to land use and climate. The results indicate that there is no general decline in occupancy across the UK, yet for northern England and Scotland occupancy declined. Very interesting are the findings that average habitat breadth declined in the south of UK whereas voltinism, the number of generations per year increased in the north. Patterns for drivers of change are mixed but indicate increased probability of presence with warmer temperatures and decreased probabilities with higher % urban area.

- Will the work be of significance to the field and related fields? How does it compare to the established literature? If the work is not original, please provide relevant references.

Comment 1A. The study is an interesting and relevant contribution to an increasing body of publications on long-term trends of insect declines, possible drivers, and interactions among climate and land use. In addition to references cited in the introduction, recent studies demonstrate temporal declines in insect biomass and abundance (Hallmann et al. 2017, Seibold et al. 2019), land use and extreme weather events as potential drivers (Seibold et al. 2019, Müller et al. 2023), and climate-land use interactions (Uhler et al. 2021, Ganuza et al. 2022). These studies might help the authors to provide a more differentiated picture of possible drivers and interactions in the introduction and discussion of own findings.

Thank you for the references, they strengthen and broaden the scope of the introduction. We have added them to our second paragraph.

LINES 65-79

We have also added a paragraph in the discussion referring to Hallman et al. 2017 (and others), when addressing the other point your raised concerning decreasing occupancy trends in the north of GB.

LINES 267-282

- Does the work support the conclusions and claims, or is additional evidence needed?

1B. In principle, yes. However, the discussion is very short and general and lacks a more causal interpretation of findings and consideration of partly conflicting other studies. I also find the introduction too short and superficial and miss the development of testable hypotheses.

We have added a finishing paragraph to the introduction in which we state clearly our hypothesis (as in e.g. Ganuza et al. 2023).

LINES 104-109

We have also expanded the first paragraph of our discussion to come back to this hypothesis.

LINES 245-256

Stating our hypothesis clarifies our objectives. However, it quickly led to scattered rephrasing throughout the *Discussion* to discuss our results under that new light. We hope you will agree it is a significant improvement to the clarity of our message.

- Are there any flaws in the data analysis, interpretation and conclusions? Do these prohibit publication or require revision?

1C. Data analysis and interpretation mainly look sounds (but see my comment below). The decline in probability of occupancy in northern England and Scotland is actually contrary to expectations, given the positive impact of temperature and the decadal increase in temperature, but this is not discussed.

We have added a paragraph to the discussion addressing this and drawing comparison to Hallman and colleagues' study of 2017, as well as other European studies that illustrates the complexity of poleward range expansion. These help in anchoring our results in a wider scientific consensus on climate impacts.

LINES 267-282

- Is the methodology sound? Does the work meet the expected standards in your field?

1D. The authors use an artificial neural networks approach to explore the high number of samples, traits and potential drivers. Without own experience with this method I can't judge how valid the analyses are, and it is difficult to reconstruct which and how analytical steps have been done by the AI. Common shortcomings of such data sets such as spatial and temporal bias or gaps in sampling data and collinearity of environmental factors have been addressed in the data analyses but it remains unclear how critical these aspects are and to which extent they might question the quality of presented findings.

We have expanded our final section *Addressing Sampling Bias* with more comparisons to alternative approaches for dealing with the inherent bias of such data sets. It now makes a stronger case for our study and explains its modelling strengths better.

LINES 509-513 ; 525-529 ; 538-540

Also, Appendices B and C have been added to provide more information about model training (Fig. S3) and performances (Fig. S2). Appendix B addresses, among other things, the collinearity that you mention here by showing the predictive performance of the models when their inputs are significantly reduced to the set of drivers highlighted by SHAP (Fig. 3).

LINES 817-872

- Is there enough detail provided in the methods for the work to be reproduced?

1E. No, I also missed basic information on the total number of occupancy data for each of the nine taxa and how these data are distributed in time and space. Most occurrence, trait and driver data are under embargo according to the authors and will be not published with this manuscript which is unfortunate.

We have added a supplementary figure (Fig. S1) presenting the sample distribution for every insect group considered here. It also features sample size per group and decades. We have added another figure (Fig. S2) that presents, among other things, the model performance

per insect group. We have also added a schematic view (Fig. S3) of the ensemble modelling approach that can help towards reproducibility as well.

LINES 809-872

We understand that the data not being readily available is frustrating aspect of this research. This is an unfortunate side effect of assembling a data set from so many different recording schemes, which we see as a strength of our study. However, we happily share what is ours to share towards reproducibility, that is the moth data.

Note that we made the code available on GitLab and the bulk of the occupancy data can be downloaded on the NBN atlas portal. Trait data bases on the other hand, are rarely readily available. We provide one for the moths that has been collated by our colleague Dan Blumgart.

LINES 559-563

References

- Ganuza, C., Redlich, S., Uhler, J., Tobisch, C., Rojas-Botero, S., Peters, M.K., Zhang, J., Benjamin, C.S., Englmeier, J., Ewald, J., Fricke, U., Haensel, M., Kollmann, J., et al. (2022). Interactive effects of climate and land use on pollinator diversity differ among taxa and scales. *Science Advances*. 8, eabm9359. <https://doi.org/10.1126/sciadv.abm9359>
- Hallmann, C.A., Sorg, M., Jongejans, E. Et al. (2017). More than 75 percent decline over 27 years in total flying insect biomass in protected areas. *PlosOne*, <https://doi.org/10.1371/journal.pone.0185809>
- Müller, J., Hothorn, T., Yuan, Y., Seibold, S., Mitesser, O., Rothacher, J., Freund, J., Wild, C., Wolz, M., & Menzel, A. (2023). Weather explains the decline and rise of insect biomass over 34 years. *Nature*. <https://doi.org/10.1038/s41586-023-06402-z>
- Seibold, S., Gossner, M. M., Simons, N. K., Bluethgen, N., Müller, J., Ambarli, D., Ammer, C., Bauhus, J., Fischer, M., Habel, J. C., Linsenmair, K. E., Nauss, T., Penone, C., Prati, D., Schall, P., Schulze, E.-D., Vogt, J. et al. (2019). Arthropod decline in grasslands and forests is associated with landscape-level drivers. *Nature*, 574. <https://doi.org/10.1038/s41586-019-1684-3>
- Uhler, J., Redlich, S., Zhang, J., Hothorn, T., Tobisch, C., Ewald, J., Thorn, S., Seibold, S., Mitesser, O., Morini, J., & al. (2021). Relationship of insect biomass and richness with land use along a climate gradient. *Nature Communications*, 12. <https://www.nature.com/articles/s41467-021-26181->

Reviewer #2 (Remarks to the Author):

This is my first time reviewing "Trait mediation explains decadal distributional shifts for a wide range of insect taxa" submitted by Bourhis et al. to *Nature Communications*. The text regards distributional shifts and the decline of insect species in Great Britain. Specifically, the authors considered 1252 insect occupancies in Great Britain in three decades from nine insect groups. The authors indicate known drivers of biodiversity loss to insect species based on trait mediation. The authors also considered species traits to determine the potential change of insect data across decades. The authors analyzed general variables affecting the amount of data for each group of species. For instance, the authors found that urbanization variables negatively affected the insect data while temperature average positively affected the species. In general, the text is well written and below I provide some suggestions of improvement that need to be considered before the text is accepted for publication.

Comment 2A. L60-63: Please note that "wider" and "higher" require a complete comparison. To what are you comparing "wider" and "higher" here? (e.g., "wider than XXX" and "higher than YYY").

We have rephrased that first paragraph to avoid incomplete comparisons.

LINES 62-64

2B. L84: Please avoid using possessives in academic texts. It is too colloquial.

Done.

LINE 89

2C. L91: "nine" - Please note that numerals lower than 10 need to be written in full length. On the contrary, numerals higher than 10 need to be written as numerals. Still, please avoid starting a sentence with a raw numeral.

Done.

LINE 96

2D. L112: Please use en dashes in numerical ranges and "one year"

Done.

LINE 128

2E. L118: Please explain the acronym "GB" here. Does it stand for "Great Britain"? This is the first time it appears without any other explanation.

When GB appears for the first time, we write Great Britain in full.

LINE 91

2F. L127: Please avoid using possessives in academic texts. It is too colloquial.

Changed to "*along the length of GB*".

LINE 143

2G. L190: Please use en dashes in numerical ranges.

Done.

LINE 206

2H. L201: Please avoid using possessives in academic texts. It is too colloquial.

Change to "Effect on the predicted probability of presence".

LINE 218

2I. L209-210: "1 km x 1 km"

Done.

LINE 226

2J. L261: Please avoid using possessives in academic texts. It is too colloquial.

Changed to "*the hostplant type of the species*".

LINE 302

2K. L299: Please use en dashes in numerical ranges.

Done.

LINE 341

2L. L316: Please note that if you are using the American English variant, you need to use a serial comma before "2020". Of you are using the British English variant, then you do not need it. Still, there are other instances where you used serial commas, for instance, L324.

Indeed, line 324 was mistaken so we changed it to the British English variant, making it consistent with the rest of the text.

LINES 362 and 370

2M. L335: Please use commas after "here"

Done.

LINE 385

2N. L341: Please avoid using possessives in academic texts. It is too colloquial.

Changed to "*the individual trait values of the species*".

LINES 392-393

2O. L348-349: Please invert this sentence.

Done.

LINES 400-401

2P. L353: American version (serial comma required) or British English (no serial comma required)?

Fixed.

LINE 405

2Q. L385: "We used the Shapley Additive exPlanations (SHAP) method"

Done.

LINE 457

2R. L389 and 394: "n = 50"

Done.

LINES 461 and 466

2S. L396: Please avoid using possessives in academic texts. It is too colloquial.

Changed to "*linear approximation of the effect of the driver*".

LINE 468

2T. L405: American version (serial comma required) or British English (no serial comma required)?

Fixed.

LINES 476 and 477

2U. L411: Please use en dashes in numerical ranges.

Done.

LINE 483

Reviewer #3 (Remarks to the Author):

In this manuscript, the authors devised a deep neural network model for species distribution modelling (SDM) and predicted distributions of British insects by training it with large scale occurrence records and trait databases. The neural network model had a specific architecture to incorporate trait mediations (trait-environment interaction) terms on top of the basic prediction model with environmental variables. The authors tested the effects of trait mediation by using the model and a variable importance measure developed on the SHAP value, a popular metric for machine learning model explanation.

The most methodological parts of this manuscript were presented in Bourhis et al. 2023 (published in *Method in Ecology and Evolution*). This study is an application of the method to broader datasets. Having read through the 2023 paper, I find most concerns and questions I had during the review have been already addressed. The models and training procedures are carefully designed and tested, and I do not find serious flaws in the methods such as data leakage. General concerns associated with SDM studies like class imbalance and biases of presence-only data were also properly addressed in this manuscript.

Therefore, the main results, like local decline of insects and the significant interactions between voltinism and temperature, are convincing. I feel that this study is a good application of the explainable AI, which can not only predict the species distributions but also detect important drivers of the decline.

Comment 3A. Only one methodological aspect I think to be added is the test of importance measures. If I understand it correctly, the SHAP-based importance measure used in this study is a novel method proposed in the 2023 paper, and it has been tested only with the butterfly and moth data sets. The same testing procedure like removing unimportant features and checking performance reduction may strengthen the method's applicability.

We have added a supplementary figure (S2) and section (appendix B) showing the performance drop from halving the number of drivers after selecting only the ones highlighted in Figure 3. This additional section helps answering questions regarding data leakage, hyperparameter tuning and the relevance of SHAP driver highlights. We agree that it strengthen our case and helps towards reproducibility.

LINES 817-856

3B. Apart from the methodology, one problem of this manuscript is it lacks detailed descriptions of the methods used. The current method section is not sufficient for reproducing the study. Probably, the method section requires more specific details of the models, training procedure and whether there are differences between this study and the 2023 paper. For example, Bourhis et al. 2023 proposed and tested multiple architectures and training procedures. Which ones were used in this study?

We hope you will agree that the aforementioned (comment 3A.) new supplementary section alleviates some of your concerns regarding reproducibility. We have also added which architecture from our 2023 paper was used throughout the study: "The approach uses ensemble of two ANNs whose predictions are averaged (i.e. the *ensemble* architecture presented in Bourhis et al. 2023⁴⁸).", as well as in the schematic Fig. S3 (see comment 3C).

LINES 390, 864-872

3C. In addition to more specific descriptions of models, schematic figures like the ones in the 2023 paper may help readers to capture the overview of the model.

We have added a schematic view of the ensembling process (Fig. S3) in the supplementary sections. It builds on the 2023 schematic figure, and recall which of the tested architecture, namely *ensembling*, was used here. Unfortunately, we believe it is difficult to bring this schematic into the main text, but strongly agree that having it available will help readers that are curious about the methodology. The supplementary figure S3 is referred to in *modelling* section in *Material and Method*, when explaining the ensembling process

LINES 413, 864-872

Minor points:

3D. Line 100: This study's results strongly depend on the performance of the model's prediction. Please briefly report the model's performance metrics, maybe MCC?, early in the result section.

We believe it would affect the flow of the argument to introduce model performance metric at this point in the results section. Quite a bit of information is needed for the MCC to make sense here. However, we agree that model performance should be mentioned at this point, so we direct the readers towards Fig. S2 (and the supplementary section B) for a summary of model performances. This way, when predictions are first mentioned, the reader is reminded that they are contingent on model performance, as you have suggested.

LINES 123-124

3E. Line 199: The effects of insect body size may act on the absolute values and may be erased when the relative size class is used to aggregate the effects on multiple taxa.

It is a very plausible interpretation, notably in all that regards dispersal abilities. Were we to have a consistent measure of insect size, like simply the average length of an individual, we could then investigate it. Unfortunately, merely because of trait data availability, we have to rely on trait measurements that are often only relevant to their specific insect group: e.g. the length of the tibia for bees, the length of the forewing for butterflies, the length of the whole individual for carabids, etc. This imposes the within-group scaling that we have also applied to the 2 other cross-group traits.

However, because of the incredibly diverse anatomy of insects, having measurements of the same nature for all insect (maybe like body size or wing area) would not necessarily solve this issue. For example, it is not the same organ that drives (or relates to) dispersal from one group to the next, so by using an insect-wide standardised measurement we could miss trait mediations that build on dispersal abilities. Finding group-relevant metrics that broadly characterise an ecological feature, such as dispersal or reproducing strategies, as we did here, is therefore more likely offer an insect-wide overview of trait mediations.

To address this idea, we have added "*Note that because we lack trait measurements of the same nature across insect groups, especially for insect sizes, we do not investigate their absolute insect-wide effects. Instead, those measurements are scaled within each group (see Fig. S4) so that we can highlight within-group effects that are consistent at the wider insect level.*" in the Material and Methods section after introducing the traits.

LINES 376-380

3F. Line 366: How many locations is "a set number of random locations"?

That number is varying and explained further in the following sentence. That number is defined per year and insect group to equate the (test) sample size for that year and that insect group. This is to ensure comparable (epistemic) uncertainty estimates between predictions and observations. To alleviate the confusion, we have decided to rephrase those 2 sentences to:

"Then, to circumvent the sampling bias from the data, we investigate the predicted trends at random locations across GB. To produce comparable uncertainties for the observed and predicted trends, the number of random locations is set per insect group and year to match the number of samples in the (test) data set for that year and insect group."

LINES 433-437

3G. Line 419 onward: I am generally convinced by the arguments here and Appendix D, but this section lacks some contexts. Can you briefly describe the common approaches used to correct the biases in the SDM applications, and how the current approach compares to them?

We have added mentions to alternative practices aimed at alleviating those usual concerns in SDMs. It helps connecting our approach to existing practices the reader might be familiar with.

LINES 508-513, 525-529, 538-540

3H. Line 451: What type of weighting is used in this study? Since the weighting strategy is critical to the model's good performance, more specific information should be provided. Can you write it down as an equation?

The weights used here are the `sklearn.utils.class_weight.compute_class_weight()` from the `scikit learn` library. It balances the classes according following formula:
$$n_samples / (n_classes * np.bincount(y))$$
so that if we have e.g. 4 absences and 2 presences, the absences get a weight of 0.75 and the presences a weight of 1.5. We have found this is an effective weighting for presence-only data. However, for presence-absence data, best performance was obtained by softening the weighing scheme by taking its square root (i.e. for the example above, weights are now 1.22 and 0.87).

We have added a paragraph in the Method section that presents those weights. They are also addressed in the supplementary section B (Fig. S2), in which the selection of the weight exponent (linear: 1, square root: 0.5) is illustrated.

LINES 414-427, 834-835

3I. Gillespie et al. 2024 proposed a loss function which could control the similar biases by introducing a different weighting scheme. There may be some commonality between two approaches. (<https://www.pnas.org/doi/10.1073/pnas.2318296121>)

Thank you for a very interesting read. My understanding of Gillespie's *sampling-aware binary cross-entropy* loss function (following their supplementary description) is that it is equivalent to ours, that is a binary cross-entropy with linear class weights. In our case, as in theirs, the class weights are species-specific, which is a technicality demanded by the nature of multispecies SDMs (this was also the case in our 2023 paper). We have added a reference to Gillespie's study when introducing the use of species-specific class weights.

LINES 350, 533-534

3J. Line 721 (Appendix D): Is this simulation based on previous studies? If so, please cite them.

This experiment builds on the literature related to the 3 fixes described in section *Addressing sampling biases* (with citations appearing then). Those *fixes* are bagging, species-specific

class weights, and within-group presences as pseudo-absences. Building on their synergetic interaction is a novelty from this study (and of our previous 2023 paper).

Reviewer #3 (Remarks on code availability):

3K. The repository contains a brief instruction and all codes required for reproducing this study, including models, training procedures and visualization. You can reproduce the workflow with a decent computer and a typical environment for running deep learning models. I could run the code on my workstation but could not on my laptop because of high resource requirement.

Thank you for having given it a try. The data set that we are at liberty to share, i.e. the moth data, is the one with the largest number of species. Running it demands a lot of memory. I have updated the repository with a more recent version of the minimal code in which we can subset the species. I have provided a simpler way to try it with ones' own data, as well as to subset the moth data to a smaller number of species. Now the whole prediction/explanation/prediction can be tested in a matter of minutes if you try it on 20 moth species for example. We think this could help towards the wider appropriation of our developments and I will continue to update the minimal model in that direction. Thank you again for all those very valuable inputs.

REVIEWERS' COMMENTS

Reviewer #1

I am reviewing this manuscript the second time. The revised version has taken into account my previous comments. The introduction, results and discussion are now clearer and more differentiated. I can accept that a more nuanced discussion of findings is not feasible in this format and see the value of such long-term and multi-taxa synthesis work.

The additional information on methods and modelling in the supplementary material is very helpful.

Reviewer #2

This is my second time reviewing "Trait mediation explains decadal distributional shifts for a wide range of insect taxa" submitted by Bourhis et al. to Nature Communications. The first draft was already pretty decent and I see no major problems with this new improved version. After Reading it again, I raised some very minor issues shown below.

L34: "(e.g. butterflies)"...

We would like to argue against this change. These taxa were chosen based on the availability of trait data and sufficient species observation coverage. As such, they represent well-known groups that both expert and non-expert readers are likely to recognize and relate to.

L43: "landscape diversity and voltinism" (no semi-colon).

Done.

Please check whether the journal requires the use of keywords at the end of your abstract (or not).

No keywords needed.

General input to your intro: Considering the suggestion made by Reviewer #1 regarding the inclusion of some key references (e.g. Hallman et al. 2017), I believe you need to include other key references in your main text (intro and discussion). They are directly related to the topics presented in your study.

Thank you for those very relevant references indeed.

Dirzo, R., Young, H. S., Galetti, M., Ceballos, G., Isaac, N. J. B., & Collen, B. (2014). Defaunation in the Anthropocene. *Science*, 345, 401–406.

Added line 64.

<https://www.science.org/doi/10.1126/science.adp4671>

Added line 61.

Regarding landscape change, this new study is a must-be-cited one: <https://www.nature.com/articles/s41586-025-08688-7>

Added lines 71 and 284.

Regarding climate change, the study by Pecl et al. 2017 needs to be cited as well. Pecl, G. T., Araújo, M. B., Bell, J. D., Blanchard, J., Bonebrake, T. C., Chen, I.-C., Clark, T. D., Colwell, R. K., Danielsen, F., Evengård, B., Falconi, L., Ferrier, S., Frusher, S., Garcia, R. A., Griffis, R. B., Hobday, A. J., Janion-Scheepers, C., Jarzyna, M. A., Jennings, S., ... Williams, S. E. (2017). Biodiversity redistribution under climate change: Impacts on ecosystems and human well-being. *Science*, 355, eaai9214. <https://doi.org/10.1126/science.aai9214>

Added line 64.

L97: "groups (e.g. butterflies,)"

See answer to comment L34 above.

L102: "mapped"

As you suggest below, we have changed to past tense all actions leading to the present discussion of the paper. This includes data manipulation, model training and predictions (i.e. most verbs in the manuscript). To contrast with those resolved actions, we would like to keep a minority of verbs in the present tense for observations such as trends in the original data or in the figures, as those remain observable to this day, as well as for the discussion.

L110: please remove this line.

Done.

L114: "translated"

See comment L102 above.

L117: "were used to train"

See comment L102 above.

L120: "resulted" (please use the simple past tense throughout the entire Results section, as this tense is often indicated to be used in this kind of section).

See comment L102 above.

L184, L185, and L186: "(e.g. ladybugs" and "(e.g. carabids and bees)" and "e.g. temperature", respectively (since you appear to be using the British English variant in your text).

Done.

Figure #5: Could you consider a different color palette here? Blue and green are pretty close and difficult to see. I think there are R packages capable of allowing you to color your figures in more easy-to-see colors. Please check this (at least for Fig5A).

We have changed the colour palette for Figure 5, leading to clearer contrasts but maintaining coherence between Figures 5A and 5B for drivers that are found in both subplots.

L245: "showed" (please also use the simple past tense in all verbs related to the results you are presenting in the Discussion section).

See comment L102 above.

L323: Please avoid possessives in academic texts. It is too colloquial.

Done.

L340-341: Considering that there are vast available biological information for these groups in online datasets like GBIF and other museums and considering the baseline shift hypotheses you proposed as one of the explanations of your results in the discussion, please explain to your readers why you did not use GBIF data with dates prior to 1990 here.

Added lines 331-333, the main reason is that high quality land-cover data in the UK starts in 1990 and that, given the importance of land-covers in our study, building on

composite land-cover information would necessarily results in artefactual discontinuities in the predicted outputs.

For the discussion, since you used iRecord and iNaturalist data, I think it would be interesting to call your readers' attention regarding the importance of citizen science to help develop analyses like ours.

Indeed. We have added a new start to our conclusion that highlights the importance of citizen science data to our study, line 285-288.

Please use the simple past tense in your M&M section as well, when you explain the methods you used to reach determined results.

See comment L102 above.

L369: "species (e.g. 58 butterflies,)"...

See answer to comment L34 above.

L408: Please, but please avoid using the "to get" verb and all of its variants in academic texts. It is too colloquial!

Changed to "to become".

Reviewer #3 (Remarks to the Author)

This manuscript is a revised version of the manuscript I reviewed. It reports insect distribution shifts and their major drivers predicted by a neural network model trained with long-term occurrence datasets.

Authors thoroughly responded the points raised by the reviewers. Additional texts and figures are informative, especially, Fig S1 and S2 provide details of model evaluation and model overviews. Also, the difference between this work and Bourhis et al. 2023 is clearer in the current version. I think now the manuscript and codes provide sufficient details for ensuring evaluation and reproduction of this research. As I already mentioned in the initial review, the manuscript's results are convincing and now they are better discussed.

Overall, I think the manuscript is now in very good quality and ready for publication.

There is only one minor point to be added:

Line 524: how about adding non-abbreviated form ("bootstrap aggregating") after "bagging"?

Yes, we have added it, lines 494-495.

(Remarks on code availability)

It is the same repository as the previous review, but with a smaller dataset for better accessibility. All codes for analyses and visualisation are included.